# A Design Framework of Exploration, Segmentation, Navigation, and Instruction (ESNI) for the Lifecycle of Intelligent Mobile Agents as a Method for Mapping an Unknown Built Environment

**DOI:** 10.3390/s22176615

**Published:** 2022-09-01

**Authors:** Junchi Chu, Xueyun Tang, Xiwei Shen

**Affiliations:** 1School of Architecture, University of Nevada, Las Vegas, NV 89154, USA; 2Department of Computer Science, Brown University, Providence, RI 02912, USA; 3Interior Architecture Department, Rhode Island School of Design, Providence, RI 02903, USA

**Keywords:** artificial intelligence, autonomous agent, unknown built environment, hierarchical framework, path finding, robotic system design

## Abstract

Recent work on intelligent agents is a popular topic among the artificial intelligence community and robotic system design. The complexity of designing a framework as a guide for intelligent agents in an unknown built environment suggests a pressing need for the development of autonomous agents. However, most of the existing intelligent mobile agent design focus on the achievement of agent’s specific practicality and ignore the systematic integration. Furthermore, there are only few studies focus on how the agent can utilize the information collected in unknown build environment to produce a learning pipeline for fundamental task prototype. The hierarchical framework is a combination of different individual modules that support a type of functionality by applying algorithms and each module is sequentially connected as a prerequisite for the next module. The proposed framework proved the effectiveness of ESNI system integration in the experiment section by evaluating the results in the testing environment. By a series of comparative simulations, the agent can quickly build the knowledge representation of the unknown environment, plan the actions accordingly, and perform some basic tasks sequentially. In addition, we discussed some common failures and limitations of the proposed framework.

## 1. Introduction

Intelligent agent development theory is a widely discussed topic in the AI community. Its impact on several industries as a core driver for cutting-edge technologies is considered significant. Russel and Norvig [1] claim an agent is anything that can be viewed as perceiving its environment through sensors and sensors acting upon that environment through actuators. The agent has intelligent behaviors, mathematically an agent function, to specify the choice of actions by perceiving the surrounding environment. Franklin [2] offers a taxonomy of autonomous agents through various properties: environment, sensing capabilities, actions, drives, and action selection architecture. Hence, this research examines how an agent’s awareness and perception of its current environment influence the analysis of its knowledge system. This research considers the agent has an objective function as a goal to achieve, the measurement of how well an agent can execute human commands to maximize the expected value of the object function. A system integration reveals the value of designing a sequence of modules to help intelligent agents to complete goals.

In comparison to the current literature on intelligent agent design [3,4,5,6,7,8], the proposed ESNI model indicates alternative benefits and suggests its potential: first, the agent can learn knowledge without supervision and is highly autonomous to acquire information without significant effort from users. Second, the agent proves its autonomous states by utilizing its explored information to build a novel knowledge system.

The main contribution of this research is to design a novel framework by using ESNI system integration for an embodied agent for use in a residential plan in a real-world situation (See Figure 1). This agent can learn the surroundings without supervision, and autonomously build its knowledge based on the algorithm integrated into the system. In this innovative framework, this agent can understand human commands and perform simple actions given its navigation trajectory and can mobilize freely with a grid environment. In general, this research discusses a novel approach for autonomous service robots to explore and navigate unknown environments by experimenting within a typical residential grid-oriented plan environment.

Several applications of intelligent mobile agent in robotics have proved the importance of agent’s functionality and utilization, whereas the adaptiveness of an agent in different environment has not been extensively studied. The current research focuses on realistic industrial AI application in known or pre-defined environment. For example, the dialog intelligent agent has a specific function of communicating. A surgical robot deals with surgical procedures by using robotics systems. Unfortunately, neither of these raise a research question about how the intelligent agent investigates the unfamiliar settings, acquires useful information from the unknown build environment, learns the knowledge representations by processing and analyzing the data collected. The research question for the paper is how the intelligent agent builds a knowledge representation system to utilize for basic task.

## 2. Literature Review

### 2.1. Intelligent Agents in Unknown Environments

This section focuses on the literature covering the theoretical study of intelligent mobile agents’ property in unknown environments during the past 10 years and presents an overview of the current state-of-art methods for intelligent mobile agents utilized in four different modules.

According to the International Organization for Standardization (ISO 8373) [9], intelligent robots require “a degree of autonomy”, and involve the “ability to perform intended tasks based on current state and sensing, without human intervention”. If there is no human activity involved, the machine can be considered intelligent when it makes decisions based on the interpretation of its recognition from the unknown environment. Furthermore, with a powerful computational resource and the ability to perform deep analytics, intelligent agent should extend human’s cognition while dealing with the complex uncertainty and equivocality, instead of simply replacing human’s contribution, Jarrahi [10] enhanced that.

The literature review is summarized respectively for 4 ESNI stages. The most recent methods in each stage are compared and discussed in Section 2.2. Given the combination of the existing methods, a knowledge gap that the lack of connections between how the agent utilizes information in unknown build environment and the achievement of basic task is discovered and thus it is necessary to propose a novel theoretical framework in Section 2.3.

#### 2.1.1. Exploration

Intelligent agent exploration tasks involve a level of uncertainty that is typically constrained by ways to acquire information in an unknown environment. Most of these exploration methods have been classified into two categories: unsupervised exploration [11,12,13,14,15,16] and supervised exploration [17,18,19,20,21,22]. For the former, Mirco [11] casts the exploration problems to an entropy maximization equation, induced by reinforcement learning algorithms. They develop “alphaMEPOL” as a policy gradient algorithm to optimize the results and empirically discussed the performance of algorithms in continuous environments. Niroui [12] proposed a novel “partially observable Markov decision process” POMDP approach that divided exploration into subtasks with a well-defined reward function in a time-sensitive unknown environment for rescuing. Fickinger [13] provided 2 policies (Explore and Control) adversarial agents and outperformed the current state-of-art unsupervised exploration methods in exploration and zero-shot transfer tasks.

Raul Mur-Artal [14] worked on the development of exploration theory based on SLAM, he upgraded the original version to ORB-SLAM, a feature-based SLAM, allows wide baseline loop closing and re-localization, and includes fully automatic initialization. For supervised exploration methods, Kaushik [17] presented a model-free policy-based approach called Exploration from Demonstration (EfD); where human supervision is necessary to provide guidance for search space exploration. Sudo [18] proposed a whiteboard model as an intermediate information storage that allows the autonomous agents to study the location information at each node in a distributed structure.

#### 2.1.2. Segmentation

Grid map segmentation is a method to convert undefined places into meaningful semantic sub-areas, and traditional methods mainly focused on divisions of floor plans into individual grid units to ease computational efforts. Bormann [23] made a survey of a collection of the current segmentation methods: morphological segmentation, distance transform-Based segmentation, Voronoi graph-based segmentation, and feature-based segmentation. However, all those methods encounter a common problem of instability with smaller area and lower compactness. Markus [24] proposed an approach of using occupancy grid mapping to derive instance based semantic maps by utilizing an improved method where occupancy grid mapping adapts to dynamic environments by applying an offline and an online technique to learn the parameters from agent’s observations. Saarinen [25] combined the OGM method with an independent Markov Chain, a two-states transition to have representations that encoded occupancy of the cells in a dynamic environment. Fermin-Leon [26] proposed an alternative approach called contour-based topological segmentation by finding the exact convex decomposition of the contour instead of depending on the disarticulation of grid cells.

#### 2.1.3. Navigation

At a minimum, the autonomous mobile agent’s basic functionality should incorporate a high-level navigation technology. Two major approaches are dealt with: traditional navigation methods and A* based navigation methods. In the former, the agents are expected to acquire information of surrounding landmarks and update their position as a recognition of self-localization to ensure the correctness of navigation. To successfully navigate to any destination assigned, a path-finding algorithm is essential, thus Jiaoyang [27] propose a new framework rolling-horizon collision resolution (RHCR) by applying multi-agent path finding (MAPF) solver to avoid collision between agents. The paper extended the research into simulated warehouse instance and successfully outperformed the existing approaches. Quoc Huy [28] discussed safe path finding by using a combination of three techniques: particle filter, Bézier curves, and support vector machine. Frantisek [29] takes advantage of the modified A* algorithm, an algorithm that is composed of a heuristic estimation function and a cost function, for the purpose of searching for the shortest path while avoiding obstacles. Zhang [30] promoted an A* Dijkstra integration method to avoid collision and deadlocks, in which they select the optimal output from the result of these two algorithms. Gang [31] proposed a geometric A* algorithm applied to automated guided vehicle (AGV) by setting up the filter functions to eliminate invalid nodes that cause irregular paths, with a focus on solving issues on many nodes, long distance and large turning angles.

#### 2.1.4. Instruction

A key bottleneck in deployments of robots is for users to provide commands for intelligent agents, since training an agent to process open-ended natural language is challenging. Jayant Krishnamurthy [32] introduced logical semantics with perception (LSP), an experimental model for grounded language acquisition that maps two physical objects in two applications: scene understanding and geographical question answering. Hatori [33] designed an integration of neural network architectures including SSD (image chopping), CNN (image classification), LSTM (language processing), and MLP (prediction), which takes images and languages as input to resolve ambiguity in spoken instructions; and he claims this model can deal with physical objects with unconstrained natural language instructions. Thao [34] found deficiencies in the above two models and developed a method that requires the verb to specify a task, and proposed an intermediate transition deemed as contextual query, information represented with a restricted format that maps specific commands to agents. Eric [35] proposed a second intermediate transition based on the usage of the contextual query to simplify the command; a refined structure called lifted linear temporal logic. The twohop transition reduced the complexity of mapping and proved to be efficient for agents to receive instructions from humans by mapping from the functional CQ to specific tasks.

### 2.2. Method Comparison

The limitations discussed above indicate a research gap in understanding different perspectives for intelligent agents. In the overall analysis of methods, the current literature on exploration indicates over-exploration, wandering issues, and high dependency on human supervision. Over-exploration means these methods are restricted in spatially limited areas; and hence the overview map of the environment can be construed as partially correct. Wandering issues create additional computational time and appear to reduce the efficiency of the agent’s exploration process. Human supervision is reliable for correcting labels but losing the meaning of automatic exploration in an unknown environment. Furthermore, it is a paradox that human supervision is accountable in an unknown environment due to lack of prior data.

In terms of segmentation, most approaches suffer from low accuracy due to the inability to capture the variance from dynamic environments. In dynamic environments, the features surrounding the agent may indicate homogeneous changes and cause unpredictability. Another point is the importance of the known robot pose in the OGM method. The agent’s map construction would be incorrect in the OGM method. The classic OGM method must convert and categorize all coordinates as either filled or unfilled. However, identifying a semi-transparent object, such as the glass is problematic and requires a non-binary representation. There is no OGM-related method that clusters a group of points or defines a group of points as a boundary to define a spatial region and hence, the categorization of areas in the unknown environment is unable to be identified. Researchers have considered navigation methods. However, some challenges appear: simulation environments are not able to represent the real-world applications; and determining an optimal path during mobility is challenged by physical obstacles while minimizing the traversal cost. The classic A* algorithm is widely used in the intelligent agent’s navigation system, but some drawbacks emerge: the heuristic value determines the performance of A* algorithm. A* is not able to operate efficiently within dynamic environments and with the operating assumption that every action has a fixed cost, this would not be verifiable in a real world. Additionally, the direct application of the A* algorithm to find paths within the structured environment is computationally high.

To deliver effective commands to the agents, the current state-of-arts reveals these drawbacks: disconnected relationships in terms of logic, poor performance due to unseen vocabularies, and limited availability of task types. Natural languages have rules to express thoughts, but not necessarily in a well-formed structure. Using machines to capture the pattern of languages is challenging, and it is even harder with the issue of ambiguity. However, to behave in an “intelligent” way means an intelligent agent must have the confirmation of actions map to the expression, so the significance of finding an accurate transition between spoken languages to specified tasks brings the topic worth discussion.

### 2.3. Theoretical Framework

Given the apparent lack of research to integrate the four methods indicated earlier to design an intelligent agent, this research provides an opportunity to experiment and formulate a new framework (See Figure 2). Instead of considering the methods as discrete individual modules, this work examines the potential derived by the inter-connections and tangents among the respective methods. This research seeks to contribute to the emergent body of knowledge on experimental work and applied research in AI, especially robotics as autonomous entities as dynamic agents within the studies of the built environment. The proposed novel framework expands the discourse on robot design as active agents to gather data in the built environment, has the potential for systematizing robot design, influences the process of autonomous agents, and becomes a systematic and standardized robot design. The anticipation of another research will focus on the trade-off of local optimization for each module and aims to achieve global optimization. Another possible direction is to extend other perspectives to improve the efficiency of functionality or the diversity of functionality types. Second, for application wise, the framework maps from a working environment to an unknown home-type environment, thus having a significant impact on the promotion of barrier-free communication agents to provide end-to-end service to users. It reduces the complexity and redundancy of communication between machines and humans and is committed to improving human-computer interaction experience. The proposed models and algorithms will contribute to the development and progress in intelligent agent theoretical study and guide the implementation for promising real-world application.

### 2.4. Key Terms in Proposed Method

#### 2.4.1. POMDP

Partially observable Markov decision process (POMDP) [36] is a stochastic process that describes the discrete states for embodied agents in an environment. Formally, Markov decision process (MDP) is represented by a tuple, (S, A, T, R, γ), where S is all possible states, action domain is the set of all possible actions, transition function T = T(s, a, s′) = Probability(s’|s, a) is a set of conditional probabilities between states, R is a reward function, and γ is the discount factor. In MDP, all states are known to the agent to make an optimal solution, but in POMDP, the agent only has an observation O to relate the states. The tuple representing POMDP has additional two elements, O and Z: the observations space O is the set of all possible observations and Z is the set of conditional observation probabilities [37].

#### 2.4.2. Contextual Query Language

The contextual query language is a formal language that retrieves information in a well-defined structure [38]. The query, in general, must be human-readable and writable while maintaining the featured information of complex original languages. The meaning of using contextual query in this environment is to map from a border generalization of human instructions to a selection of a well-defined task. More specifically, the verb and noun words are extracted as keywords parameters in the model.

#### 2.4.3. Occupancy Grid Mapping

The occupancy grid mapping refers to a representation that the continuous space in the environment was partitioned into different cells. Each cell has a Bayesian probability to indicate whether it is 1 (occupied) or 0 (empty). The representation is simply the situation of mobile robots and has become the dominant paradigm for embodied agent environments [39]. Occupancy grid mapping (OGM) method is used in module: segmentation to convert the simulation into grid map.

#### 2.4.4. A* Algorithm

A* Algorithm is a graph traversal and pathfinding computer algorithm that is widely used in games development. The advantage of A* algorithm is that it can search the shortest path from source to destination with hindrances. A* algorithm selects the path with a minimum cost function f(n) = g(n) + h(n), in which g(n) is the cost from the source node to the current node, and h(n) is a heuristic function that estimates from the current point to the destination point. The selection of heuristic functions depends on the problem, but in general, Manhattan distance or Euclidean distance are the two most popular heuristic functions to use [40].

## 3. Research Method

### 3.1. Jargon

Symbols: Before elaborating on methodologies to address the tasks in this project, important symbols and terminology definitions are given as follows. The collections of all moveable objects in the grid world are denoted as M = {$m_{1},m_{2},m_{3}$ …,$m_{n}$}. The targeted grid’s coordinates are denoted as (${\mathrm{gird}}_{x},{\;\mathrm{grid}}_{y}$) and the target object mj’s coordinates are denoted as (${\mathrm{mj}}_{x}$, ${\mathrm{mj}}_{y}$). The agent’s current location is denoted as (${\mathrm{cur}}_{x}$, ${\mathrm{cur}}_{y}$).

#### 3.1.1. Sections

The collection of sections is denoted as: S = {$s_{1}$, $s_{2}$, …, $s_{n}$}. For each grid cell c, a section $s_{j}$, j ≤ x is an accumulated grid cell cluster, c∈S, that can represent the functionality of a house, for example, kitchen, living room, bathroom, etc.

#### 3.1.2. Boundary Points

The collections of points lay between two different sections. For any point that is identified as boundary points $b_{n}$, there exists a neighbor point that belongs to a section that is different from bn’s section. Mathematically, it supposes a boundary point (${\mathrm{bn}}_{x},{\;\mathrm{bn}}_{y}$), then $\forall \;\left({\mathrm{bn}}_{x},{\;\mathrm{bn}}_{y}\right)\in$$s_{p},\;\exists \left({\mathrm{bn}}_{x}+a,{\mathrm{bn}}_{y}+b\right)\in s_{q},\;\mathrm{such}\;\mathrm{that}\;a,b\;\in \left\{-1,0,1\right\},\;a+b\;\in \left\{-1,1\right\}\;\mathrm{and}\;p\neq q$.

#### 3.1.3. Section-Wise Path

A section path is from one section to another section. The purpose of section pathfinding is to find a path at high levels to help the agent’s navigation trajectory. For example, to navigate a point from n kitchen to the bathroom, the section path is Kitchen → Studio → Bedroom → Bathroom. The section path can be generated by the BFS algorithm discussed in the section segmentation paragraph.

#### 3.1.4. Walking Values in Exploration

Walking value is a term to describe the frequency of the agent visit on a particular grid. A dictionary that stores the information of each grid to associate a walking value is created with initiation of all walking values to 0. The program increases the value by 1 if and only if the agent visits that position.

#### 3.1.5. Bin Values

The highest bin value determines the belonging of a grid cell. Each grid cell has six bins since there are six room types in the simulation. The formula to calculate bin value is provided in Section 3.4.

#### 3.1.6. Ground Truth Knowledge Dictionary

Knowledge dictionary is a data structure that stores key value pairs to indicate the belongings of each object. For example, [TV] = {Bedroom, Living Room}, [Chair] = {Kitchen, Living Room, Studio}, [Bed] = {Bedroom}.

#### 3.1.7. Maximum Allowed Steps (MAS)

This terminology is used in the experiment part to describe how many steps the agent is allowed to explore the unknown environment. In the program, the agent’s movement is suspended when the maximum allowed steps is reached.

#### 3.1.8. Percentage of Occupied Cells (POC)

This terminology is to describe how many space the agent has occupied or covered in the experiment. The percentage of occupied cells range from 0% to 100%.

### 3.2. Agent Model and Set Up

As Figure 3, this research assumes that the agent has no prior information of a floor plan. The computer program converted the architectural floor plans into the digital 40 × 40 grid world by using the OGM method. Moreover, it defined a white area as a walkable space for the agent. Any space other than white color is non-walkable. The initial position of the agent is located at the right corner of the building, with no prior knowledge of the environment. When initializing the environment, objects are instantiated in the zones by sampling the attribute distributions in the knowledge dictionary, which captures an ontology of locations and objects in the building. Objects are classified as key fixture objects (bedroom), non-key fixture objects (sofa, table, and TV bench, etc.), and movable items (such as apple, bowls, chairs, etc.).

### 3.3. Exploration

The goal in this step is for the agent to cover maximum unvisited spaces, identify all the boundary points, and build the unweighted connectivity graph to generate a pathfinding algorithm. The algorithm (See Algorithm 1) can be described in two steps: initially, a dictionary WALKING_VALUES can map from all positional and walkable grids to a numerical value. Then the program marks a grid that has been visited by increasing the value by 1 if the agent has stepped over or passed that grid. In this simulated environment, the agent has four legal options for the next move, UP, LEFT, DOWN, RIGHT. The potential choice of the next move will be removed if the agent will collide with an object or obstacles. Based on four possible move options, the agent chooses the move which has the minimum walking value in the dictionary. Intuitively, a move directs the agent to a grid that has a higher walking value than others, meaning that the grid is more frequently visited compared to others. The agent will choose the next move of the grid with the lowest value among the four directions.

In the two dimensional grid world, the POMDP tuple has (S, A, T, R, γ, O, Z), where S describes states as all possible coordinates in the unknown environment. “A” has four legal directions: UP, DOWN, LEFT, RIGHT. “T” is the probability transition between states, so whatever action are chosen could cause the walking values for each grid. “R” is the reward function, a positive reward can lower walking value, and no reward for staying. γ is the discount factor with a default value of 0.9. The observation “O” is partial, thus the agent has no information for any state that is not adjacent. Z stands for the conditional observation probabilities depending on “O”.

**Algorithm 1:** Agent Exploration AlgorithmAgent’s current positions (curx, cury) Initiate a dictionary: walking_valuesWhile Maximum Allowed Steps has not been reached do   Add Adjacent S: states of curx, cury to walking_values   Choose Moves <——A: [“UP”,”DOWN”,”LEFT”,”RIGHT”]   Remove move if collides with fixture objects & Update T, O, Z Loop each move in Adjacent MOVES do  Move = argmin (walking_values [move]) & Update R  Walking_values [(curx, cury)] + = 1

### 3.4. Section Segmentation

The agent uses a section segmentation model that guesses the area correspondingly to distinguish the different functionalities of the areas (such as balcony, studio, kitchen, etc.) for every single grid cell. After agents explore all spaces in the grid world, all item information must be recorded and the next procedure for the agent is to predict areas for each grid. The agent uses an area prediction algorithm to determine which area a grid belongs to.

The environment is divided into six basic types according to the functions of the general home space: living room, bedroom, bathroom, studio, kitchen, and balcony. All elements in the simulation are classified into three categories, key objects, non-key objects, and moveable items (See Table 1). A key fixture object refers to a signature object that determines the functionality of a section, for example, the bathroom has a toilet as a key object and a bed must be in the bedroom. The simulation has three key fixture objects, the toilet, bed, and gas cooker that can be regarded as the key fixture objects for deciding the room types. However, for the other three types, lack of key fixture objects requires us to use other objects to measure the space types. Non-key fixture objects refer to an object that increases the probability of a grid cell to be assigned as a section, for example, TV as a common non-key fixture object can be found in living room and bedroom. Moveable item refers to any item that can appear in any room during daily usage, for example, it would not be surprising to find a teacup in the living room or bedroom.

Applying the above classification principles, the agent assigns each grid a room type based on the probability of what functionality this room should be, by calculating the surrounding objects. There are a few restrictions to be noticed. First, each object will contribute a non-negative bin value v to grid cell c. v is determined by the reciprocal of Euclidean distance times the factor N. A higher Euclidean distance results in a smaller bin value contribution. Second, for any grid cell c, object o does not contribute bin values if and only if a path between c and o exists wall objects. Third, any moveable item or non-fixture object has a contribution of 0 bin value.

The overview of the area prediction algorithms is described as follows (See Algorithm 2). For each grid, 6 sections are associated with 6 possible assignments, thus corresponding to 6 bins. For every single grid cell, the agent calculates the Euclidean distance between each object and the grid cell and adds up the weighted contribution value to the corresponding bin for each item associated with the bin. The contribution value is defined as the reciprocal of Euclidean distance by times with a factor N, to the associated bin to account for how possibly this grid belongs to the section. The default N = 50 for key objects, N = 1 for non-key objects, and N = 0 for moveable items. The bin that has the highest value determines the ownership of the grid cell. Mathematically, the bin calculation formula can be represented as the following:(1)$$\mathrm{Bin}\_\mathrm{Value}=\sum_{j=1}^{j=n}\frac{1}{N\left({\left(grid_{x}-mj_{x}\right)}^{2}+{\left(grid_{y}-mj_{y}\right)}^{2}\right)}$$


(2)
$$N=\{\begin{array}{cc} 50 & if\;Key\;Fixture\;Objects\\ 1 & if\;Non-Key\;\mathrm{Fixture}\;Objects\\ 0 & if\;Moveable\;Item \end{array}$$


**Algorithm 2:** Agent Section Segmentation AlgorithmWhile existing a grid without section segmentation do   Initiate a histogram: Hist_Area that has 6 bins: balcony, living room, bedroom, bathroom, kitchen, studio, and bedroom   For each object in grid world do     Calculate Bin_Value     Loop for each bin in Hist_Area do       If bin in Ground Truth Knowledge[o] then         Bin += Bin_Value     Area = argmax (Hist_Area)

### 3.5. Navigation Trajectory

In the grid world, exploration helps to build unweighted connectivity (Figure 4) graphs to instruct the agent to detect a path from its current location to the destination point in the section-wise levels. A deterministic algorithm (Breadth-First Search) [41] can determine the shortest path from the current section to the destination section in a high level, for example, Kitchen→ Studio→ Bedroom→ Bathroom. The agent uses the section segmentation function to locate its current section and destination section. The navigation strategy for the agent is to move to the boundary points from the current section to the next section’s boundary points in the section-wise path until reaching the destination.

For example, consider a simulation of virtual environment in a bedroom, a task requires the agent to generate a two-hop navigation trajectory from bedroom to balcony. In the high levels (section-wise), the navigation task can be split into four sub-tas: from bedroom, to kitchen, studio, and balcony. In the low level, which means for each navigation sub-task within a particular section, there could be many objects appearing in the section as obstacles. To find the shortest path to avoid obstacles, A* algorithm can navigate in the low-level. The A* algorithm selects the path that minimizes f(n) = g(n) + h(n), where n represents the next points, g(n) is the cost of the path from the boundary point to n, and h(n) is a heuristic function that estimates the cost of the cheapest path from n to the next boundary point. Figure 5 demonstrates how the agent navigates from the kitchen to the balcony, the blue arrows are the navigation path and red bars are boundary points between sections.

### 3.6. From NLP to Contextual Query

CQ extracts the keywords from the original language and discards unrelated information to ensure the efficiency of information retrieval. CQ can serve as an intermediate transition from natural language to machine-readable code. For example, the agent cannot have a clear map of what task it needs to complete when a human says: “I haven’t eaten since last night, I am hungry. Can you give me an apple?” The agent executes the following CQ: bring the apple to the living room. A template CQ filled with parameters will map a specific task, thus the agent understands what type of task is being requested. Table 2 lists examples from NLP→ CQ→ Tasks. CQ can be parameterized to sub-tasks, so the component of a single CQ has two parts, the template and the parameter. For the sake of simplicity, four fundamental CQ templates are defined as follows: Bring (parameter 1) to (parameter 2), Navigate (parameter 1), Find (parameter 1), and Swap (parameter 1). The last CQ template has no parameters and can be used as a default status to maintain the static for the agent. Each class of contextual query corresponds with a task parameterized by a goal. NLTK [42] as a pre-build package can tokenize and tag the queries, then analyze the sentence composition to extract the goal parameters. The goal is to implement custom transformation mapping from each contextual query class to goal-parameterized tasks.

## 4. Experiment

The experiment demonstrates the trade-off between steps taken by the agent and the sections he can explore in the grid world. The grid world simulation has limitations in terms of dimensionality and size. Theoretically, partial exploration of the environment always results in inaccurate segmentation. The maximum allowed step (MAS) is passed as an input for each iteration. The agent stops the exploration and performs the section segmentation once the maximum allowed steps are reached.

### 4.1. Exploration Experiment

For the exploration experiment, the agent starts at MAS of 50, and the computer program increases MAS by 50 each time, until 2000, to find an optimal MAS point where the percentage of occupied cells (POC) converges. The average results of 20 iterations are taken for each epoch of MAS.

### 4.2. Segmentation Experiment

For the section segmentation experiment, the same measurement metric was used as the exploration experiment, to get the results of section segmentation in different values of POC. Information in the previous stage is utilized for this stage.

### 4.3. Navigation Experiments

The computer program generates different results of navigation trajectories, given the information of segmentation stage. In this simulation, testing in 2000 random pairs of 2 points and comparing the correct label path with the generated navigation trajectories by the intelligent mobile agent are carried out.

### 4.4. Instruction Experiment

Testing on different inputs under various contexts reveals that the language does not have any pattern, but the NLTK model still extracts the accurate information as parameters to put in the function prototype. 2000 short human commands are tested in this experiment, and the result is measured based on how accurately the agent can capture the keywords and identify the correct task prototype in the dialogue, by comparing with the correct label of the parameters.

## 5. Results

### 5.1. Results for Exploration

When the maximum allowed steps are relatively low, the agent hovers around in those areas near the starting point. If MAS was increased, the agent should ultimately step in other sections that he has not previously covered. A percentage of covered space with respect to the maximum allowed steps can demonstrate the result of exploration. Figure 6 shows the percentage that the agent has covered or explored, given the maximum allowed steps. To achieve approximately 80% explored space in the grid cells, the agent has a MAS of 750. The curve converges to 100%, which means that the agent has fully explored all the grids in this environment when the maximum allowed steps are around 1600 steps.

### 5.2. Results for Section Segmentation

The number of sections the agent can predict accurately is determined by the number of cells that the agent can occupy. Theoretically, all sections are segmented well if the agent can fully explore the grid cells and observe all objects in the environment. Figure 7 demonstrates the segmentation results at different POC, based on the average results of 20 experiments. The grid cells that the agent segments with the ground truth label and the error of segmentation as the ratio of incorrect labels of segmentation/total grid cells in the simulation were compared. Figure 8 shows the relationship between POC and error of segmentation. The error of segmentation is 69% when POC is 10%, which means the agent recognizes most of the grid cells as part of kitchen. When POC is 30%, the agent can reach to most of grid cells in the living room and can recognize the living room cells correctly. The agent can recognize 3–4 sections correctly when POC is 50%, and only mark one section wrong (mostly the bathroom was incorrectly labelled since it is the latest places to be visited). The agent’s error rate reduces to 3% if POC is close to 90%.

### 5.3. Results for Generating Navigation Trajectories

Navigation trajectories vary in the result of segmentation, thus also have dependency on the MAS. In the simulation, with a high segmentation error rate, the section connectivity unweighted graph missed component of the grid world. Based on 2000 trails of random pair point’s navigation, the system bears a certain degree of segmentation error rate, that is, up to 12% the agent maintains the correctness of path finding. For a 50% error rate, the correctness significantly drops to 25%, which indicates that the missing component in section connectivity weighted graph is imperative for the success of generating a correct navigation trajectory (See Figure 9).

### 5.4. Results for Instruction Experiment

In this stage, testing is focused on the correctness of task prototype and capture of corresponding parameters. Even though the command inputs are short sentence, different perplexities of language affect the performance of the model. The comparison from the results of the instruction model with respect to the ground truth labelling reveals common failures that the agent cannot solve because of the ambiguity of the language. If the contextual query is missing parameters or mapping to wrong task prototype, the sequence of action cannot be generated correctly.

## 6. Discussion and Conclusions

A problem in traditional intelligent agent design is the lack of a lifecycle of how the agent can be autonomous by collecting information from the unknown environment. We introduced a framework of intelligent agent design and provided several algorithms during the pipeline for the agent to perform tasks. The framework has novel points for discovering hidden information during the section segmentation and generating a navigation trajectory. The autonomous agent processes human languages and utilizes the trajectory to perform several actions. We use different MAS to know the trade-off between explorations and exploitation. Finally, we discussed the limitations of the proposed approach and the extended research in the relative areas.

The experiment shows that ESNI system integration successfully achieve the goal of being autonomous by sequentially executing each stage one by one. The navigation trajectories accuracy almost approaches to 100% with approximately 1300 steps of exploration. Compared to the previous frameworks, the ESNI can be used as a design framework that meshed resources together, thus the feasibility and autonomy can be ensured.

One of the limitations of this work is that the intelligent agent can provide the service under the scope of the home environment since all elements are discussed based on smart home implementation. Thus, this framework cannot be applied or need to be revised largely for other places such as metro stations, schools, or nursing homes. Moreover, our project relies on the assumption that the agent can move freely in the grid world, not considering the physics of the agent, but the agent can collide with obstacles that are never seen before in the memory. Another limitation is that the tasks type is limited due to the form of CQ being in primitive structures and the agents cannot process a more complicated dialogue with ambiguity.

The future research of the ESNI framework need to be addressed in the following three perspectives: First, the optimization problem is a big topic that needs to be researched. For each stage of the ESNI, the local optimization does not guarantee a global optimization. Second, how should we evaluate the efficiency of the framework in terms of aiming to reach a global optimization? The evaluation metrics (runtime, entropy, energy conversion rate, etc.) must be carefully selected or established from the perspective of energy saving. Last but not the least, the current format of the intelligent agent can successfully perform basic task, but still being far away from a truly autonomous agent that have a more advanced reasoning system and decision-making mechanism.

## Figures and Tables

**Figure 1 sensors-22-06615-f001:**
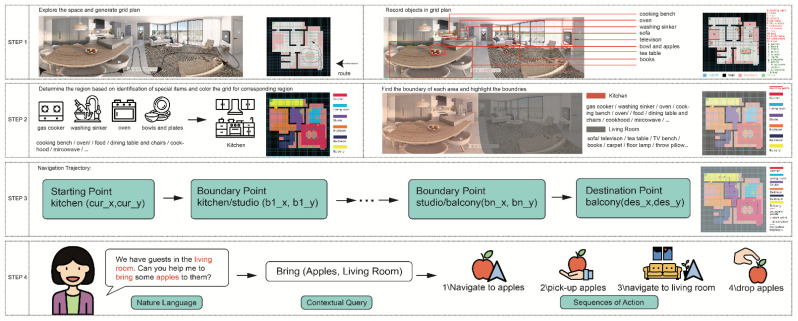
Overview of the ESNI design, step 1: maps floor plans to grid world, step 2: exploration and section segmentation to obtain all information, step 3: use boundary points to generate navigation trajectories, step 4: from NLP to contextual queries to process commands.

**Figure 2 sensors-22-06615-f002:**
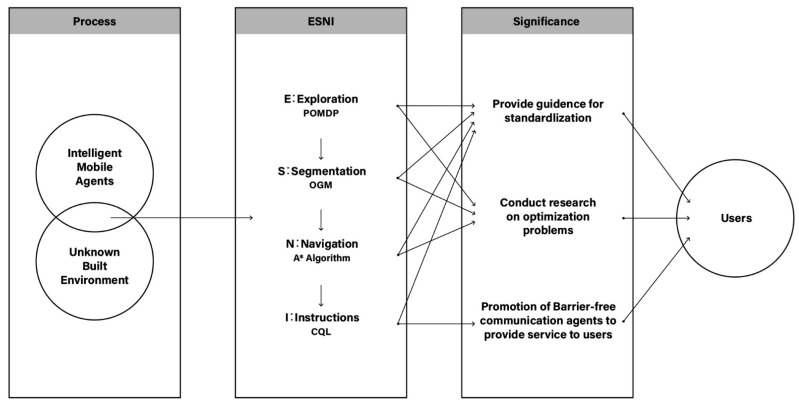
Theoretical framework.

**Figure 3 sensors-22-06615-f003:**
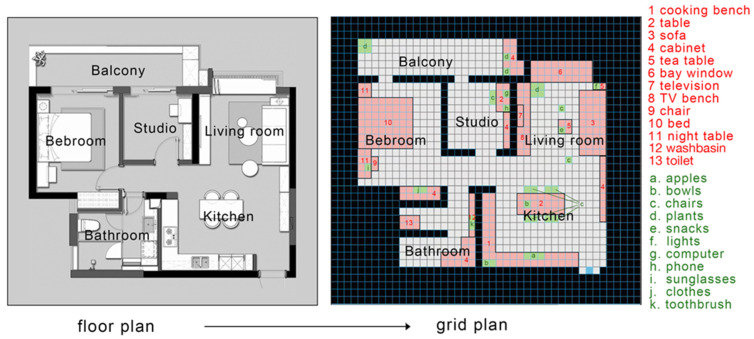
A partition from house plan to grid cells.

**Figure 4 sensors-22-06615-f004:**
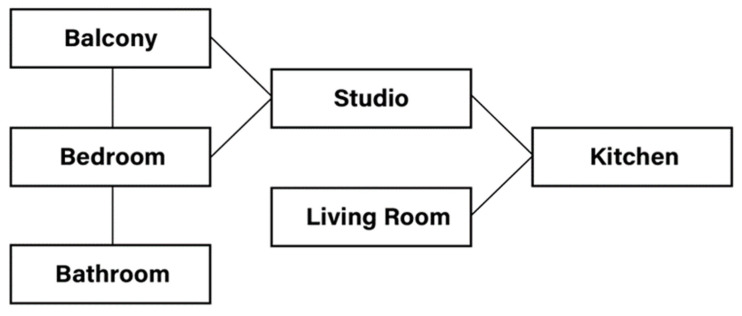
Section connectivity unweighted graph.

**Figure 5 sensors-22-06615-f005:**
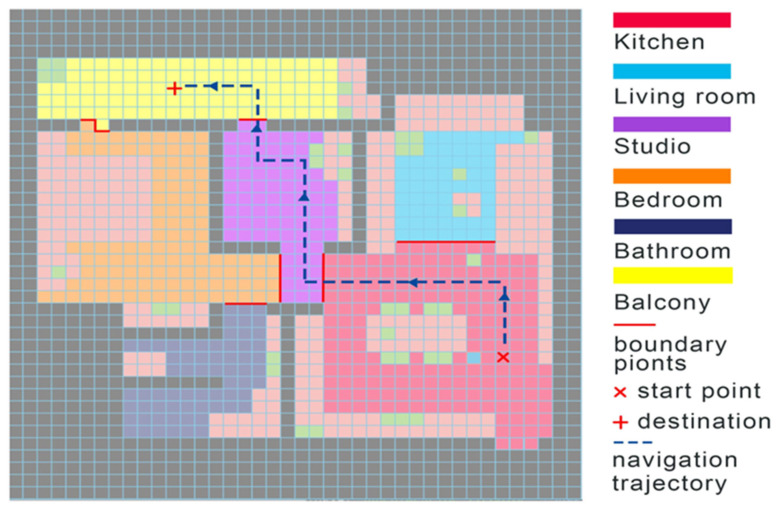
A navigation trajectory from kitchen to balcony.

**Figure 6 sensors-22-06615-f006:**
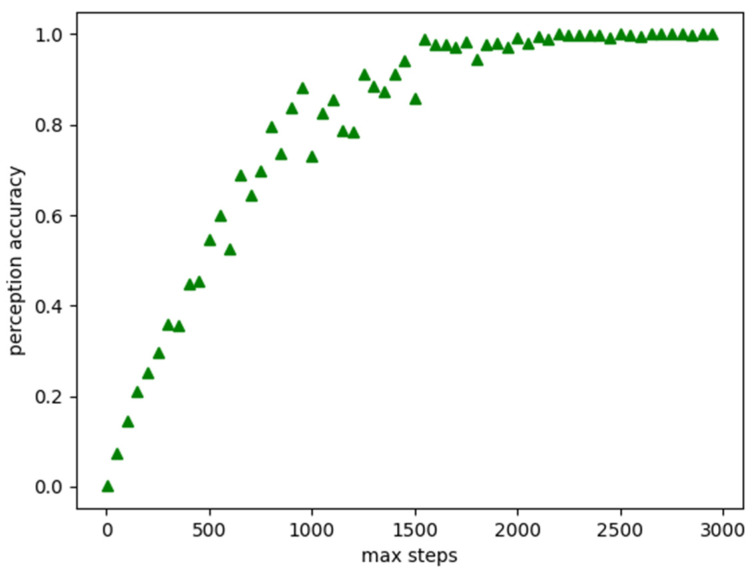
Relationship between MAS and POC.

**Figure 7 sensors-22-06615-f007:**
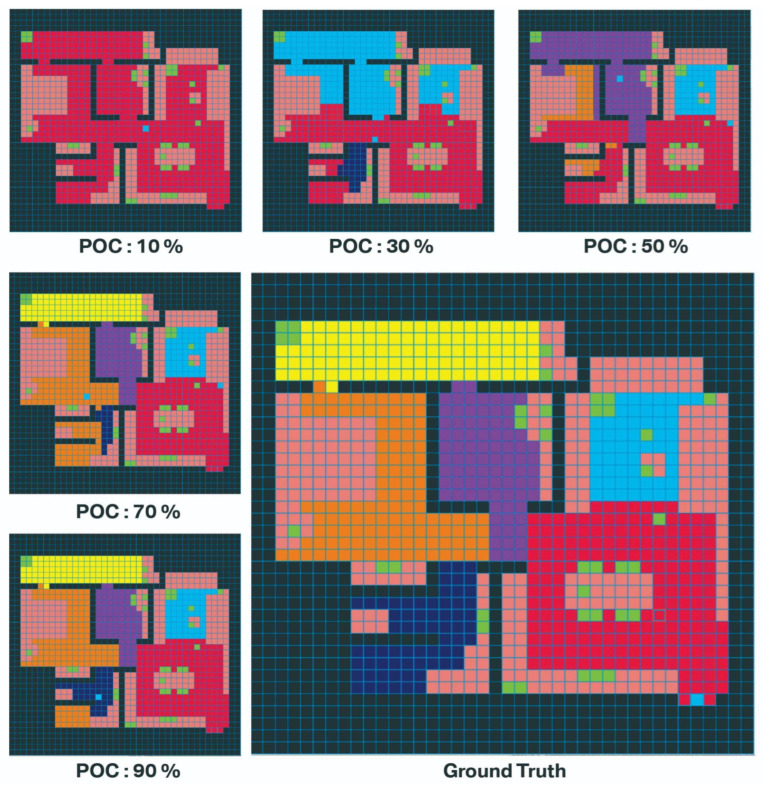
The result of section segmentation for different POC in the computer program visualization. The color blocks representation can be found at Figure 5.

**Figure 8 sensors-22-06615-f008:**
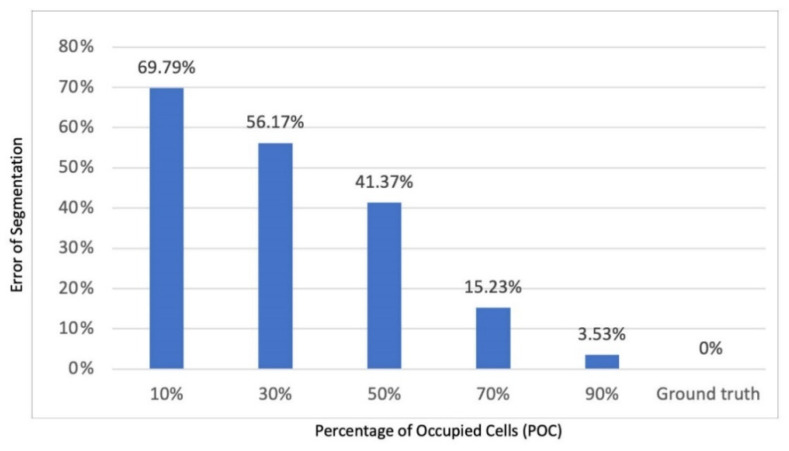
Twenty trials experiments’ section recognized on different MAS.

**Figure 9 sensors-22-06615-f009:**
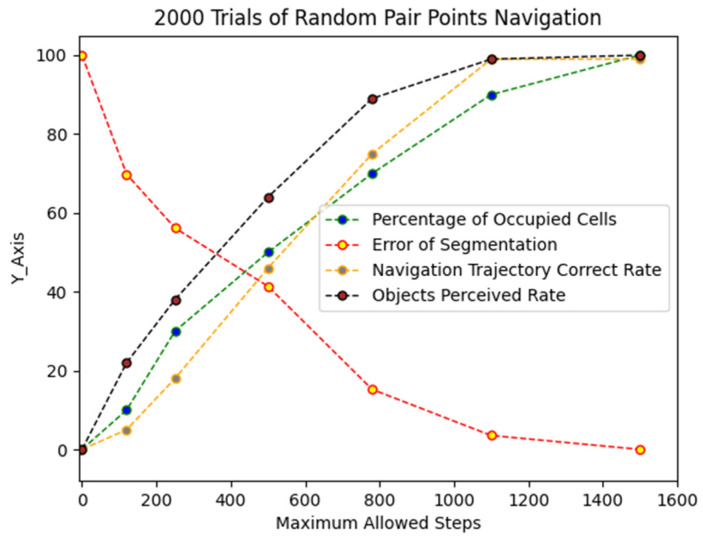
2000 trails of random pair points experiment.

**Table 1 sensors-22-06615-t001:** Groups of space types and with association of key objects.

	Section	Key Fixture Object	Non-Key Fixture Object	Moveable Item
Section with Key Fixture Object	Bedroom	Bed	Closet, Wardrobe, Night Table, Bed Lamp	Pillows, Toothbrush, Bowls, Plates, Condiment Bottles, Floor Lamp, Chair, Books, Laptop, Plants, Vase, Apple, Bananas, Pears, Oranges, Eggs, Snacks
	Bathroom	Toilet	Washing Sinker, Bathtube, Shower Head	
	Kitchen	Gas Cooker	Oven, Mircowave, Kitchen Washing Sinker, Cookhood, Refrigerator	
Section without Key Fixture Object	Living Room	N/A	Sofa, Television, Tea Table, TV Bench	
	Studio	N/A	Bookcase, Desktop PC	
	Balcony	N/A	Big Window, Curtain, Wind Chime	

**Table 2 sensors-22-06615-t002:** Examples of NLP -> CQ -> Tasks.

Examples of Mapping between NLP→CQ
Task 1: I want a banana. I am at bedroom Bring [Banana, Bedroom] Task 2: Can you come to my bedroom to serve? Navigate [Bedroom] Task 3: Hey, where is my computer? I can’t find it. Find [Computer] Task 4: Hey, I want to take a shower. Can you swap my cloth and toothbrush? Swap [Cloth, toothbrush] CQ→Sequences of Action Bring [Banana, Bedroom] = Find [Banana]→Navigate [Banana]→Pickup [Banana]→Navigate [Bedroom]→Drop [Banana) Navigate [Bedroom) = Navigate [Bedroom] Find [Computer] = Navigate [Computer] Swap [Cloth, toothbrush] = Find [Cloth]→Navigate [Cloth]→Pickup [Cloth]→Find [Toothbrush]→Navigate [Toothbrush]→Pickup [Toothbrush]→Drop [Cloth]→Drop [Toothbrush]

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
