# Peer review of "A Design Framework of Exploration, Segmentation, Navigation, and Instruction (ESNI) for the Lifecycle of Intelligent Mobile Agents as a Method for Mapping an Unknown Built Environment"

_sensors, 2022, doi:10.3390/s22176615_

Round 1
Reviewer 1 Report
With a combination of different individual modules that support a type of functionality by applying algorithms, this research proposes a design framework (ESNI) that is composed of four modules. While recent work on intelligent agents is a popular topic among the Artificial Intelligence community and Robotic System Design and the complexity of designing a framework as a guide for intelligent agents is needed for the development of autonomous agents in an unknown built environment, there is no a clear explanation regarding the research gap in the manuscript. However, there are the following issues in the manuscript, which need to be further revised by the authors:
Abstract: the abstract should be revised to explain the knowledge gap and/ or research problem rather than the research process in the current format. Moreover, the contribution of the study to existing knowledge must be revealed in the abstract.
Introduction: the authors missed to clearly address a question or a problem that has not been answered by any of the existing studies or research within the research content. So, in the introduction, you must clearly state the need for this study (with a range of citations) and what has been solved in the prior studies, and what knowledge gaps remain, nonetheless.
Literature review: generally, this part was organized well, but I guess the review of the body of literature is still incomplete, and the manuscript has missed reviewing the recently published papers in the field.
Page 7: theoretical framework must be re-drawn. It is not clear the interaction between Process on ESNI and, ESNI on Significance. Do ESNI components impact Significance components independently?
Furthermore, the proposed theoretical framework must be supported by suitable references and evidence that the manuscript missed in the current format.
Conclusion: this part is essential for a high-quality paper. It is suggested that the author re-write this part. In addition, please make sure the conclusions section underscores the scientific value-added of the paper, and/or the applicability of your findings/results, as indicated previously. Please revise the conclusion part in more detail.
the conclusions must be thoroughly supported by the results presented in the article or referenced in the secondary literature. Generally, authors should enhance the manuscript's contributions, limitations, and the applicability of their findings/results and future study in this session.
Reviewer 2 Report
In this paper, the designed framework of Exploration, Segmentation, Navigation, and Instruction (ESNI) for the lifecycle of intelligent mobile agents as a method for mapping an unknown built environment was investigated. The paper shows valuable introduction, a well-described methodology and a discussion of the related research results. In general, this paper is interesting and is in the journal scope. However, some revisions are needed before publicaiton:
(1) In the abstract, the important conclusions and results including some quantitative results obtained from this study should be introduced.
(2) The section of 2. Literature Review should be revised. The literature should be summarized briefly, and the existing problems should be presented. At the end of this section, the main work conducted in this study shall be proposed based on the summary of literature review.
(3) The English grammar should be polished for the full text. The research work conducted in this study should be presented in the past tense.
(4) The quality of Figures 2, 6, and 7 should be improved.
(5) For the articles of science and technology, the first person voice shoud not be used, such as subjects of I, we. There are too many ”we” and ”our” were used.
(6) On lines 438-443, the styles of Eq. (1) and Eq. (2) are disordered.
(7) The detailed information on experiment should be added and improved.
(8) In the results and discussion, the analysis and discussion should be strengthened. The related research results from other investigators should be used to compare with the results obtained in this study.
(9) The discussion part in the section of 6. Discussion & Conclusion should be removed to the section 5. And the conclusion should be as a separate part in several summarized items.
(10) Most of the references listed at the end of the manuscript are too old. Some latest literatures in the near past three years should be added.
Round 2
Reviewer 1 Report
Dear authors,
After carefully reading the revised manuscript, I found that the revision was greatly improved. I have no more questions.
Congratulations!